# Knowledge and attitudes of antenatal mothers towards intimate partner violence in the Gambia: A cross-sectional study

**Joseph W. Jatta** [1,2]*, **Jean Claude Romaric Pingdwinde Ouedraogo**[3]

**1** Slum and Rural Health Initiative Network, Research and Collaboration Department, SRHIN/Africa, Ibadan, Nigeria, **2** Directorate of Health Research, Ministry of Health, Banjul, The Gambia, **3** Research Institute of Health Sciences (IRSS), Department of Traditional Medicine and Pharmacopeia, Pharmacy (MEPHATRA-Ph), Ouagadougou, Burkina Faso

* jattajoseph28@gmail.com

## Abstract

Intimate partner violence is a severe public health and human rights issue that 1 in 3 women experience in their lifetime. A woman's knowledge and attitudes towards intimate partner violence might influence her experience of violence from an intimate partner. This study aimed at analysing the knowledge and attitudes of antenatal mothers towards intimate partner violence. This study employed a descriptive cross-sectional technique. Pregnant women seeking antenatal care were selected from 6 public healthcare facilities in the Gambian Upper River Region (URR). We used the single proportion formula to calculate the sample size, which was 373 women. Data were entered, cleaned and analysed using SPSS version 21.Majority of the participants had good knowledge about IPV (77%). The most common intimate partner violence known to the women was denial of money to hurt her (80.2%). Only 58% of the women knew that a male partner's insistence on knowing their whereabouts at all times and expecting them to seek permission before accessing healthcare is a form of violence. Eighty-seven per centof the participants had a negative attitude towards IPV (i.e., they were not precisely against the stated forms of violence perpetrated by their partners). When asked about what would warrant them to leave their partners, 67% claimed they would never leave their partners despite facing violence. According to 36% of the respondents, women stay in abusive marriages because of their children.Despite the relatively high knowledge of pregnant women on IPV, their attitudes tell a different story, which is: acceptance of violent behaviours from their partners. More work needs to be done to sensitise women on their rights not to be violated and engage the whole society in changing the social norms unfavourable to them.

## Background

Violence against women is a central human right and public health concern across the world. The World Health Organization (WHO) estimated that about 30% of women experienced sexual and/or physical assault globally [1].

**Data Availability Statement:** All data are in the manuscript and supporting information files.

**Funding:** This study was sponsored by the Pan African University, PAULESI, University of Ibadan,

Ibadan, Nigeria to JWJ. The sponsor had no role in the design and conduct of the study, the interpretation of the findings, and the decision to write this manuscript.

When a current or former partner commits physical, sexual, or psychological abuse, it is identified as intimate partner violence (IPV) by the World Health Organisation (WHO). It also includes controlling behaviours. Intimate partner violence (IPV) is the most pervasive form of violence against women. IPV occurs worldwide, transcending all contexts and socio-economic, religious, and cultural groups [2]. Although men may be abused, women experience more violence, usually from their male partner or husband [3, 4]. From a multi-country study on women's health and domestic violence, physical violence was the most prevalent, from 13% in Japan to 61% in Peru province [2]. Violence has a negative impact on abused women's physical, mental, and reproductive health [1]. Violence is most critical when it occurs among pregnant women.

Intimate partner violence could affect negatively pregnant women and the children they are carrying, with crucial adverse health consequences. IPV could be associated with induced abortion, miscarriage and giving birth to a preterm baby [2, 5]. More worrying, the prevalence of IPV during pregnancy is high in Sub-Saharan Africa, with 15%, 33%, 39.8% and 61.8%, in South Africa, Nigeria, Ethiopia and The Gambia, respectively [6–9]. A pooled prevalence from demographic and health surveys from 2000 to 2018 in 29 Sub-Saharan countries found 41.3% (95% CI: 37.4%-45.2%) [10]. It is not well elucidated if pregnancy prevents violence because IPV was found to increase or decrease with the pregnancy, and others even reported experiencing violence for the first time during pregnancy [7, 11]. The factors leading to IPV occurrence during pregnancy are similar to those outside of pregnancy [12]. Factors like longer duration in marriage, tribe, place of residence, partner's controlling behaviours, residing in rural areas, women less than secondary education were reported to be associated with IPV during pregnancy in Sub-Saharan Africa [4, 6, 9, 10]. Besides, women experiencing intimate partner violence had lower odds of ANC service utilisation [13].

Moreover, severe IPV victims have more significant emotional distress during pregnancy and postpartum than moderate IPV victims and non victims [14]. Some pregnant women find violence normal and acceptable, facilitating the perpetration of intimate partner violence [8]. Women's perceptions and attitudes towards violence vary widely across countries, and previously abused people are likely to legitimate or accept further violence. The most significant variations of reasons justifying violence were established between traditional and rural areas and urban and industrialised settings [11].

Investigations on intimate partner violence are few in The Gambia. IPV prevalence from facility-based or population-based studies is high, 61.8% and 40%, respectively [7, 15]. Further, women knowledge and attitudes towards IPV have not been described in the Gambia. This study aimed at describing antenatal mothers' knowledge and attitudes towards IPV in the Gambia. This would help to deconstruct the violent behaviours against women in the Gambian society.

## Methods

### Ethics statement

Ethical approval was obtained from the Ethical Review Board of the College of Medicine, University of Ibadan in Nigeria. Permission to conduct this study was further sought from the Research and Publication Committee of the University of the Gambia (RePUBLIC) and The Gambia Government/Medical Research Council Joint Ethics Committee (Reference number: R018031V1.1). The study followed the National Institute of Health guidelines on research involving human subjects.

## Study design and population

A hospital-based cross-sectional study was conducted among pregnant women seeking antenatal care. The target population was all women seeking antenatal care at public health facilities in rural Gambia.

The study population was all women seeking antenatal care at 6 purposively selected public health facilities of rural Gambia, namely: Basse Health centre, Garawol Health centre, Baja Kunda Health centre, Demba Kunda Health centre, Koina Health centre and Yorobawol Health centre.

## Sample size determination

The formula to estimate a single population proportion was used to calculate the sample size. The prevalence from a previous study of IPV amongst pregnant women was used as maximum variability (P) at 61.8% (Idoko *et al.*, 2015).

$$n = \frac{(z)^2 p \times q}{(d)^2}$$

Where;
**n** = the minimum sample size
**z** = Confidence interval at 95% set us 1.96
**p** = maximum variability of the population at 61.8%. i.e. (0.618)
**q** = the difference between 1 and p
**d** = maximum error allowed (5%).

The initial sample size was approximated to 373, then adjusted at 400 women with a non-response rate of 10% with the following formula ($\frac{no}{1-NR}$). Finally, 373 women were surveyed.

## Sampling procedure

A multistage sampling technique was used to select respondents. The Upper River Region (URR) health facilities were stratified into two strata based on their functions, namely major and minor health centres.

**STAGE 1.** By simple balloting, three out of the four major health centres were selected, with each health centre receiving an average of 60 antenatal women per antenatal clinic day. The same procedure was followed for the eight remaining minor health centres, and three were eventually selected. The specific sample size was allocated to each stratum using proportion-to-size allocation.

**STAGE 2.** A sampling frame was created based on the estimated number of women who attend ANC per day in each facility. We decided purposively to survey ten women per day. The systematic random sampling technique was used to select ten women to interview per day. A sampling interval of six was determined by dividing the estimated antenatal care attendees by the required number of respondents per day. Therefore, sixty was divided by the required ten, which gives six. Balloting was again used to select the first sampling unit. Two small pieces of paper with the inscription YES and NO were offered to the first participants to enter the palpation room, and if they pick YES, they become the first to be interviewed. The following respondents followed in the systematic random sample of (1, 7, 13, etc.).

## Data collection methods and tools

The data was collected between the 4th of February and the 31st of May 2019. The questionnaire used for this survey was adapted from the WHO multi-country study on women's

health [11, 16]. The following were the adoptions included: sociodemographic characteristics of respondents and their partners, family history of violence, history of child abuse, perception of IPV and history of child abuse. In addition, questions on parental quarrelling, childhood sexual abuse and substance abuse were modified to reduce the risk of offending respondents.

The study questionnaire consisted of 6 sections, namely: sociodemographic characteristics, knowledge on intimate partner violence, experience of intimate partner violence, history of childhood abuse, attitude towards intimate partner violence and physical health consequences of intimate partner violence. This study reports on the knowledge on intimate partner violence and the attitude towards intimate partner violence.

## Data collection method

Research assistants who were proficient in at least two local languages, preferably Mandinka and Wolof, were recruited to facilitate better understanding and communication between them and research participants. The questionnaire was pretested among 40 antenatal women in two health facilities in CRR. The semi-structured questionnaires were translated into local languages with the help of a translator. Participants were approached during ANC visits and interviewed. To ensure the privacy and convenience of the participants, there was usually one woman in the room with the midwives.

The purpose of the study was explained to every selected respondent to gain their cooperation. Written informed consent was obtained from each of them in the form of a signature or a thumbprint. Face-to-face interviews were conducted using semi-structured interviewer-administered questionnaires as the data collection tools.

## Data analysis

The principal researcher collected the completed questionnaires to ensure that the answers were not altered. The research team did data sorting, coding, entry, and analysis.

Quantitative data was entered into a soft copy using MS Excel and later exported to SPSS version 22 for analysis. Descriptive statistics were used. Frequencies, proportions/percentages were computed and presented in tables and graphs to describe participants characteristics.

## Results

### Sociodemographic characteristics of respondents

Table 1 shows that out of 373 respondents, approximately 27% had ages between 20–24 years, and 26% were between ages 25–29 years. Most women had no formal education (65.7%), and 3.8% attended Arabic schools. The majority were married (97%), while about 2% indicated they were divorced. Islam was practised by 98%, while the rest were Christian practitioners or traditionalists (2%).

Eighty-seven per cent (324 women) were housewives. The distribution by tribe showed that 39% of the respondents were Serahuli, 29% were Mandinka, and 26% were Fula.

### Women's knowledge on intimate partner violence

Overall, slightly above three-quarters of the study participants had good knowledge about IPV (77%). Table 2 gives the details of twenty items related to participants' knowledge about intimate partner violence. The most common behaviours' of intimate partner violence known to the women were denial of money to hurt her (80.2%); belittling or humiliating the wife in

**Table 1. Sociodemographic characteristics of respondents.**

|  | Frequency | Percentage |
|---|---|---|
|  | (n = 373) | (%) |
| **Age Group (years)** |  |  |
| 18–19 | 59 | 15.8 |
| 20–24 | 100 | 26.8 |
| 25–29 | 98 | 26.3 |
| 30–34 | 66 | 17.7 |
| 35–39 | 24 | 6.4 |
| 40+ | 7 | 1.9 |
| Undisclosed | 19 | 5.1 |
| **Level of Education** |  |  |
| No formal | 245 | 65.7 |
| Arabic | 14 | 3.8 |
| Primary education | 72 | 19.3 |
| Secondary education | 33 | 8.8 |
| Tertiary education | 9 | 2.4 |
| **Marital Status** |  |  |
| Single | 3 | 0.8 |
| Married | 363 | 97.3 |
| Divorced | 7 | 1.9 |
| **Religion** |  |  |
| Christianity | 2 | 0.5 |
| Islam | 365 | 97.9 |
| Traditionalist | 6 | 1.6 |
| **Occupation** |  |  |
| Housewife | 324 | 86.9 |
| Paid employment | 13 | 3.5 |
| Self-employed | 36 | 9.7 |
| **Tribe** |  |  |
| Serahuli | 147 | 39.4 |
| Mandinka | 110 | 29.5 |
| Fula | 97 | 26.0 |
| Wollof | 10 | 2.7 |
| Others were | 9 | 2.4 |

front of other people (~80%); kicking her, dragging her or beating her up (~80%); threatening to use or using a gun, knife or another weapon against the wife (~80%); and refusal to let her work or do any form of business to cow her (~80%). Aspects where only a few women had shown to have enough knowledge were: a male partner insisting on knowing where his wife is at all times (58%) and expecting his wife to ask for permission before seeking healthcare for herself (58%).

The assessment of women's knowledge of each form of IPV was presented in Table 3. More than 70% of the respondents had a good understanding of the different forms of violence. Thus, 74% had good knowledge of the psychological and intimate partner violence; about 79% had good knowledge about physical intimate partner violence; about 80% had good knowledge about sexual intimate partner violence, and 77% had good knowledge about economic intimate partner violence.

**Table 2. General knowledge about intimate partner violence.**

| | Frequency (n = 373) | Percentage (%) |
|---|---|---|
| Deny her money to hurt her | 299 | 80.2 |
| Belittles or humiliates his wife/partner in front of other people | 298 | 79.9 |
| Kicked her, dragged her or beat her up | 297 | 79.6 |
| Threatened to use or used a gun, knife or other weapons against her | 297 | 79.6 |
| Physically forced her to have sexual intercourse | 297 | 79.6 |
| Choke or burn her on purpose | 295 | 79.1 |
| Refusal to let her work or do any form of business to cow her | 295 | 79.1 |
| Hit her with his or some object that could hurt her | 294 | 78.8 |
| Push her or shoved or pulled her hair | 291 | 78.0 |
| Slapped her or threw things at her could hurt her | 290 | 77.7 |
| Threatens to hurt her or someone she cares about | 284 | 76.1 |
| Insults his wife/partner and makes her feel bad about herself | 278 | 74.5 |
| Does things to scare or intimidate his wife/partner on purpose (e.g. by the way, looks at her, or by yelling or smashing things) | 278 | 74.5 |
| Gets angry if she speaks with another man | 263 | 70.5 |
| He is often suspicious that his wife/partner is unfaithful | 258 | 69.2 |
| Ignores his partner and treats her indifferently | 257 | 68.9 |
| Tries to restrict contact with her family of birth | 251 | 67.3 |
| Tries to keep his partner from seeing his friends | 228 | 61.1 |
| Insists on knowing where his partner is at all times | 217 | 58.2 |
| Expects his wife/partner to ask for his permission before seeking health care for herself | 217 | 58.2 |

## Attitude towards intimate partner violence

Overall, the aggregated attitude of the women reveals that only 13% had a positive attitude towards IPV (i.e. they were against any form of intimate partner violence). In comparison, 87% had a negative attitude towards IPV (i.e. they were not precisely against the stated forms of violence perpetrated by their partners).

Most women (89%) consented to the assertion that "a good woman does obey her husband/partner even if she disagrees with his views". The majority of the women (72.4%) strongly agreed that family problems should only be discussed with people in the family; less than half of the women (46%) consented to the statement that a man must show his wife/partner that he is the boss in his home; about 58% agreed that a woman should be able to choose her friends even if her husband/partner disapproves; up to 73% agreed with the claim that a woman must have sex with her husband/partner anytime he wants it; up to 72% agreed that others should interfere if a man beats his wife (Table 4).

## Reasons for leaving violent partners

Responses from the participants regarding why they would leave a partner subjecting them to violence, as seen in Table 5, exposed that 67% would never leave their partners despite

**Table 3. General knowledge about intimate partner violence.**

| | Poor Knowledge | Good Knowledge |
|---|---|---|
| **Psychological Intimate Partner Violence** | 97 (26.0%) | 276 (74.0%) |
| **Physical Intimate Partner Violence** | 80 (21.4%) | 293 (78.6%) |
| **Sexual Intimate Partner Violence** | 76 (20.4%) | 297 (79.6%) |
| **Economic Intimate Partner Violence** | 84 (22.5%) | 289 (77.5%) |

**Table 4. Responses on attitude towards intimate partner violence (n = 373).**

| | Strongly Disagree (%) | Disagree (%) | Don't know (%) | Agree (%) | Strongly Agree (%) |
|---|---|---|---|---|---|
| A good woman obeys her husband/partner even if she disagrees with his views | 4 (1.1) | 32 (8.6) | 5 (1.3) | 96 (25.7) | 236 (63.3) |
| Family problems should only be discussed with people in the family | 10 (2.7) | 31 (8.3) | 7 (1.9) | 55 (14.7) | 270 (72.4) |
| It is necessary for a man to show his wife/partner who is the boss in his home | 42 (11.3) | 121 (32.4) | 38 (10.2) | 97 (26.0) | 75 (20.1) |
| A woman should be able to choose her own friends even if her husband/partner disapproves | 42 (11.3) | 78 (20.9) | 38 (10.2) | 123 (33.0) | 92 (24.7) |
| It is a woman's obligation to have sex with her husband/partner anytime he wants it | 21 (5.6) | 57 (15.3) | 22 (5.9) | 130 (34.9) | 143 (38.3) |
| If a man beats his wife, others should interfere | 56 (15.0) | 34 (9.1) | 14 (3.8) | 99 (26.5) | 170 (45.6) |

facing violence from him, and 25% remarked they would leave if they could not endure anymore.

## Reasons why women stay with abusive partners

The most common reasons stated by the participants responsible for women staying with abusive or violent partners were (Fig 1): their children (36%), love for the partner (19%), calmness & desire of the woman to stay in her partner's home (11%)

## Women's suggestions to end IPV

Suggestions from the women (Fig 2) towards mitigating the occurrence of intimate partner abuse were: couples should be dialogued with and advised (29%); local authorities interfering into the couple's affairs through policies and laws (24%). Six per cent 6% suggested that men should be sensitised to treat women respectfully; 5% suggested the women should be patient and obey the husband. Other suggestions included–divorce/separation of couples, women should be empowered, and family/forced marriages should be discouraged.

# Discussion

This study described the knowledge and attitudes of antenatal mothers towards intimate partner violence in The Gambia. The overall knowledge on IPV was good for 77% of the study participants. Generally, a good knowledge was also found on the different forms of IPV: psychological (74%), physical (78.6%), sexual (79.6%) and economic (77.5%). The aggregated attitude mainly was negative for 87% and positive in only 13% of the cases. About 67.3% of the participants claimed that they could never leave their partner. In comparison, most women

**Table 5. Responses on reasons for leaving violent partners.**

| | Frequency | Percentage |
|---|---|---|
| | (n = 373) | (%) |
| **Reasons for Deciding to Leave a Partner Due to IPV** | | |
| Would never leave him | 251 | 67.3 |
| Could not endure anymore | 94 | 25.2 |
| Thrown out of home | 9 | 2.4 |
| Badly injured | 7 | 1.9 |
| Threatened or tried to kill me | 7 | 1.9 |
| Encouraged by friends | 5 | 1.3 |

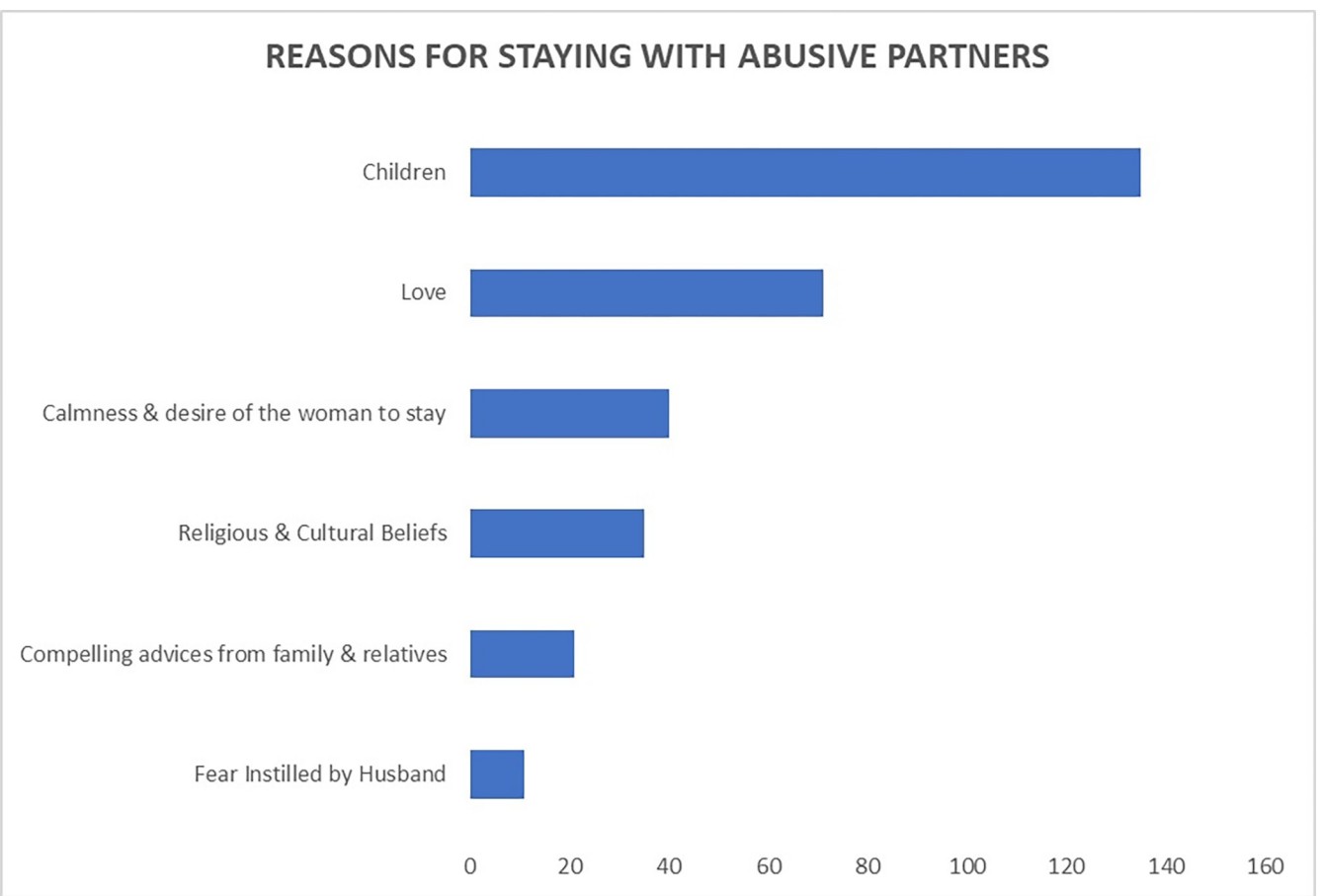

**Fig 1. Suggested reasons women stay with abusive partners.**

stayed for their children (36%) and love, religious and cultural beliefs, as well as compelling advice from family and relatives.

During pregnancy, the prevalence of intimate partner violence varies worldwide, between industrialised and developing countries [11, 17]. Further, the prevalence seems lower in population-based studies compared to clinical or hospital data. In Africa, IPV prevalence ranged from 2% to 57% during pregnancy, with a pooled prevalence of 15.23% (95% CI: 14.38%-16.08%) [18]. With these variations in IPV prevalence amongst antenatal attendees, it is likely to find differences in the level of knowledge. Antenatal care attendees showed an overall good knowledge (77%) of intimate partner violence in this study. But it was much lower than in Northwest Nigeria, where the good knowledge attained 100% [9]. Enhancing women and potential prosecutors' knowledge about violence is the first step on the road to combating violence against women. In this line, the Council of Europe Convention on Preventing and Combating Violence against Women and Domestic Violence has issued Article 13, which mandates states to raise awareness about violence [19].

The WHO considers different forms of intimate partner violence, which are: physical, sexual, and psychological violence, and controlling behaviours. The 20 questions related to knowledge were meant to assess these forms of IPV. Thus, women were also aware of the physical, psychological, sexual, and economic -taking into account some of the controlling behaviours)- violence at 70% at least. Even if physical violence is generally high, between 23% and 49% in

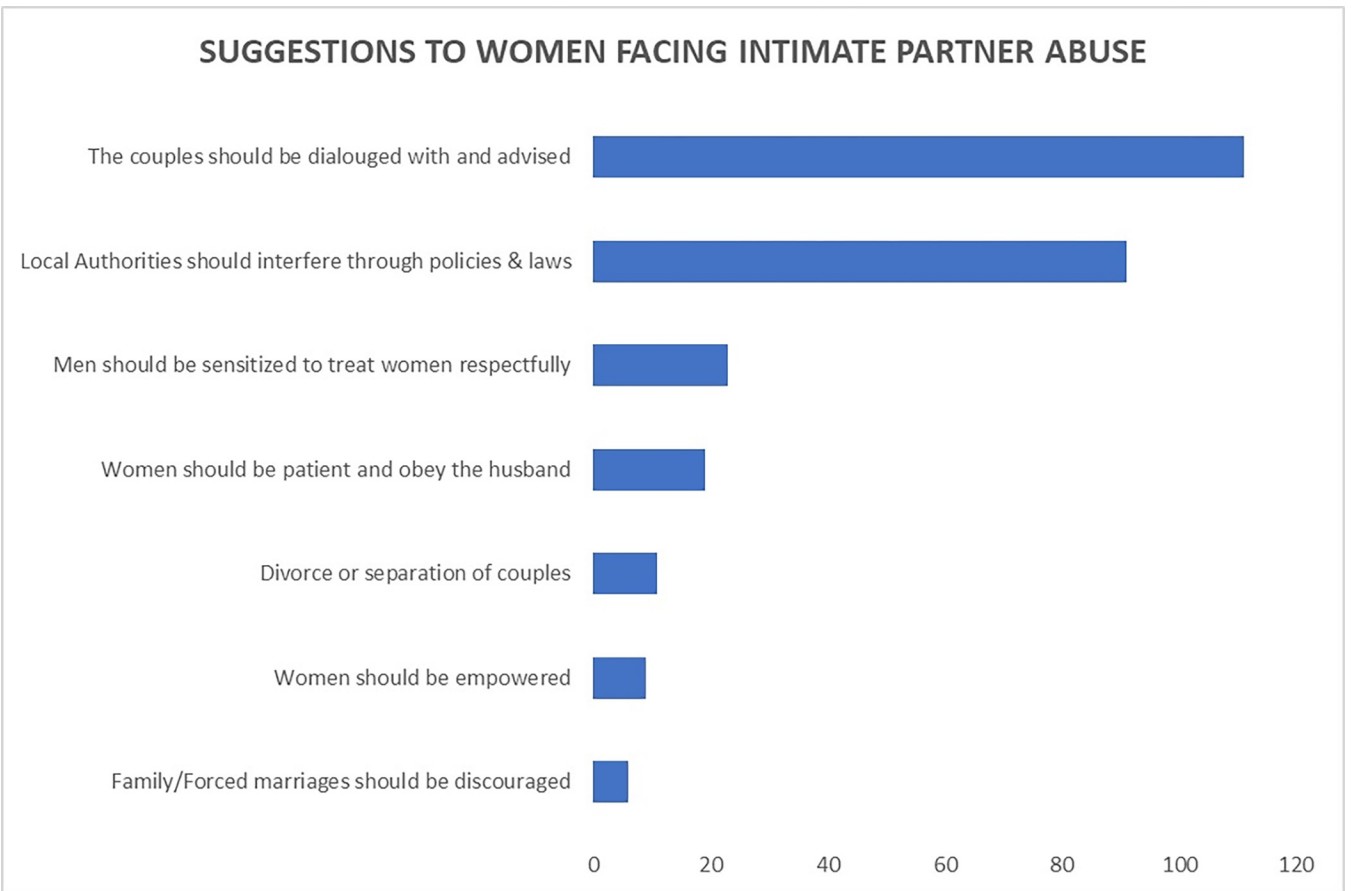

**Fig 2. Suggestions on how to end intimate partner violence.**

most sites of the WHO multi-country study, the antenatal women showed higher knowledge of sexual violence in this study [11]. Is it so because sexual intercourse is a crucial component in couples? In effect, women are at higher risk of violence from their partners than from other people [16]. It is then vital to know the situations dealing with violence, whether it is physical, sexual, or emotional. It is also essential to identify controlling behaviours in terms of restricting contact with her family of birth, keeping his partner from seeing her friends, expecting his wife to ask for permission before seeking healthcare, or restricting access to financial resources [2]. In effect, men with such behaviours are likelier to act violently against their partners [16]. Women awareness of controlling behaviours must be raised, as it felt around 58 to 69% in this study. The existence of economic pressures (77.5%) against women commands the necessity for them to be economically independent of their husbands, with income-generating activities. In The Gambia, women were primarily engaged in non-remunerable activities due to labour division between women and men [20]. To conclude, antenatal mothers' knowledge on intimate partner violence must be raised in general regarding the different forms of violence to drive good attitudes towards IPV.

Unfortunately, women had generally a negative attitude towards IPV (87%) in this study, higher than negative proportions amongst civilians (22.2%) and military women (55.4%) in Nigeria [21]. They did not show opposition to the different forms of violence perpetrated by their partners. Thus, women believed a good woman obeys her husband even if she disagrees with his views (63.3%). These proportions were higher in all groups in Nigeria, i.e. 88.0% and

85.4%, respectively, amongst women civilians and military [21]. It is said that the continuity of violence in society is guaranteed by the acceptance of violence by the victims themselves [8]. However, it could also and mostly be due to the socio-cultural context favouring the men in most African countries [22]. It could also be linked to an atmosphere of terror that maintains women under domination. Women's acceptance of violence varies worldwide, and justifications of its perpetration [11]. Acceptance of physical violence was also reported to be high amongst previously abused women [11, 15]. Few women were against the different forms of intimate partner violence (13%). It is encouraging that the tendance to absolute tolerance of violent relationships reduces, making women violence less acceptable [23, 24].

Although women experience intimate partner violence in different forms, they do not always leave the violent partner. From a multicounty study by the WHO, 19% to 51% of the women who had previously experienced physical violence never left their home [11]. Only 4% reported partner violence to the police in The Gambia, while 59% did nothing [7]. There is a general tendency for pregnant women to put up with violence from their partners. Up to 67.3% of the women reported that they would never leave the violent partner. Only the women outdone by the violence were willing to go (25.2%). Thus, staying was the general rule for antenatal mothers despite the violence, because of their children (36%), but also due to their love for the partner (19%), religious and cultural beliefs (9%) and compelling advice from family and relatives (6%). Violence installs a particular fear of the violent person in the victim, making it difficult for her to detach herself from him. The reasons mentioned above could just be pretexts to stay because of fear. In this situation, if violence does not end, women will experience it again. Because women generally stay with violent partners, it could explain why previously abused women can experience it again [25].

## Conclusion

Despite a relatively good knowledge of intimate partner violence, antenatal mothers showed a negative attitude towards IPV. Women still accept and justify intimate partner violence in The Gambia. This attitude is driven by favourable socio-cultural constraints of the Gambian society. The laws should be strengthened and implemented by the authorities, including a careful sensitisation to deconstruct such thoughts and repression of perpetrors. Behavioural change goals are long-term struggles that deserve to be made for a fairer and more equal society.

## Supporting information

**S1 Data.**
(SAV)

## Acknowledgments

Our deepest gratitude goes to the regional health team URR, the OICs and staff of all the health facilities where this study was conducted. Sincere gratitude goes to all respondents in this study.

## Author Contributions

**Conceptualization:** Joseph W. Jatta, Jean Claude Romaric Pingdwinde Ouedraogo.

**Data curation:** Joseph W. Jatta.

**Formal analysis:** Joseph W. Jatta.

**Funding acquisition:** Joseph W. Jatta.

**Investigation:** Joseph W. Jatta.

**Methodology:** Joseph W. Jatta.

**Project administration:** Joseph W. Jatta.

**Resources:** Joseph W. Jatta.

**Software:** Joseph W. Jatta.

**Supervision:** Joseph W. Jatta, Jean Claude Romaric Pingdwinde Ouedraogo.

**Validation:** Joseph W. Jatta.

**Visualization:** Joseph W. Jatta, Jean Claude Romaric Pingdwinde Ouedraogo.

**Writing – original draft:** Joseph W. Jatta, Jean Claude Romaric Pingdwinde Ouedraogo.

**Writing – review & editing:** Joseph W. Jatta, Jean Claude Romaric Pingdwinde Ouedraogo.

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
