## [Decision Letter · Decision Letter 0]

8 Nov 2022

PGPH-D-22-01560

Knowledge and Attitudes of antenatal mothers towards intimate partner violence in The Gambia: A cross-sectional study

Dear Dr. Jatta,

Thank you for submitting your manuscript to PLOS Global Public Health. After careful consideration, we feel that it has merit but does not fully meet PLOS Global Public Health’s publication criteria as it currently stands. Therefore, we invite you to submit a revised version of the manuscript that addresses the points raised during the review process.

Our second reviewer has given a detailed feedback on the paper. Kindly review them carefully and amend. In addition to our second reviewers detailed comment. I would like to draw your attention to the following. 

For a global journal like PLOS-GPH, the paper needs to bring in how the context and research is relevant for other countries (LMICs). Such an articulation is needed both in the introduction and in the discussion. I invite you to add the relevance of the research to global perspective or LMICs in addition to the detailed comments received from second reviewer. Kindly keep in mind while revising the paper, how the paper could contribute for global readers.

We look forward to receiving your revised manuscript.

Kind regards,

Muthusamy Sivakami

Academic Editor

Journal Requirements:

1. In the online submission form, you indicated that "Data is available from the corresponding author upon reasonable request". All PLOS journals now require all data underlying the findings described in their manuscript to be freely available to other researchers, either 1. In a public repository, 2. Within the manuscript itself, or 3. Uploaded as supplementary information.

2. Please upload all main figures as separate Figure files in .tif or .eps format only and remove the embedded figures from the manuscript file. For more information about how to convert and format your figure files please see our guidelines:

Additional Editor Comments (if provided):

Dear Authors,

The paper has a potential to contribute to a very important body of work. However, for a global journal like PLOS-GPH, the paper needs bring in how the context is relevant for other countries (LMICs). Such an articulation is needed both in the introduction and in the discussion. I invite you to add the relevance of the research to global perspective or LMICs in addition to the detailed comments received from second reviewer. Kindly keep in mind while revising the paper, how the paper can contribute for global readers.

Reviewers' comments:

Reviewer's Responses to Questions

**Comments to the Author**

1. Does this manuscript meet PLOS Global Public Health’s publication criteria? Is the manuscript technically sound, and do the data support the conclusions? The manuscript must describe methodologically and ethically rigorous research with conclusions that are appropriately drawn based on the data presented.

Reviewer #1: Yes

Reviewer #2: Partly

2. Has the statistical analysis been performed appropriately and rigorously?

Reviewer #1: Yes

Reviewer #2: Yes

3. Have the authors made all data underlying the findings in their manuscript fully available (please refer to the Data Availability Statement at the start of the manuscript PDF file)?

Reviewer #1: Yes

Reviewer #2: Yes

4. Is the manuscript presented in an intelligible fashion and written in standard English?

Reviewer #1: Yes

Reviewer #2: Yes

5. Review Comments to the Author

Reviewer #1: The paper main claim is significant to to the discipline. The study is suitable for publication in its current. The manuscript is well organized and written clearly to be accessible to non-specialists

Reviewer #2: Knowledge and Attitudes of antenatal mothers towards intimate partner violence in The Gambia: A cross-sectional study

This is an interesting study of an under-researched population, describing women’s prevailing attitudes and knowledge levels of intimate partner violence when seeking antenatal care in The Gambia. However, it is missing a great deal of detail in the Methods section, which needs to be strengthened. Below are some comments that may be helpful to the authors.

Abstract: The abstract is missing some crucial information in the methods section, including use of validated questionnaires and participation rates.

Line 49 – Could “issue” be missing from human rights?

Line 75 – Please spell out ANC.

Line 89 – Missing reference.

Line 88 – This sentence is a little unclear.

Line 93 – Do you know what percentage of antenatal women utilize the public health facilities in URR? Do their demographics differ from the general antenatal population?

Line 95 – “Purposively selected”, what was the reasoning behind this choice?

Line 110 – I don’t quite understand this sentence, did you ask 400 women to participate with 373 agreeing to participate?

Line 120 – Was the turnover similar in major and minor health centres?

Line 127 – At what stage and how was the study introduced to the possible participants?

Line 128 – Who offered the pieces of paper? The research assistants or health care personnel?

Line 132 – I think it would be helpful to show some sample questions and possible answers, i.e. are these yes/no answers? Any missing, and if yes how was that handled?

Line 149 - Please spell out CRR.

Line 198 - I think there is an extra and “psychological and intimate partner violence”.

Line 203 – I don’t believe that this dichotomy of good vs. poor knowledge was introduced in the methods section.

Line 218 – I don’t believe that the Likert scale is mentioned in the methods section. Also missing in the methods section is how positive and negative attitudes are defined.

Line 224 – More detail on this table in the methods section would be appreciated.

Line 234 – I don’t believe the information presented in Figure 2 was introduced in the methods section.

Line 238 – Please describe how many women raised these other suggestions.

Line 260 – Is the Nigerian study comparable to this study?

Line 267 – This sentence is a little unclear.

Line 269 – Are you referring to the prevalence or knowledge levels in the WHO study?

Line 282 – Missing reference.

Line 286 – Please describe this Nigerian study. Are these groups comparable?

Line 297 – This sentence is a little unclear.

Line 302 – Where does this statistic come from?

Line 303 – Missing reference.

Line 304 – Did you mean undone rather than outdone?

Line 308 – Missing reference.

Line 316 – 320 – The conclusion is too sweeping considering that this was one descriptive study on knowledge and attitudes of antenatal women.

A very minor issue - there is a lack of consistency in the choice of when to use capital letters, both in headings and when discussing The Gambia.

6. PLOS authors have the option to publish the peer review history of their article (what does this mean?). If published, this will include your full peer review and any attached files.

**Do you want your identity to be public for this peer review?** For information about this choice, including consent withdrawal, please see our Privacy Policy.

Reviewer #1: No

Reviewer #2: No

---

## [Editor Report · Decision Letter 1]

20 Dec 2023

Knowledge and Attitudes of antenatal mothers towards intimate partner violence in The Gambia: A cross-sectional study

PGPH-D-22-01560R1

Dear Mr Jatta,

We are pleased to inform you that your manuscript 'Knowledge and Attitudes of antenatal mothers towards intimate partner violence in The Gambia: A cross-sectional study' has been provisionally accepted for publication in PLOS Global Public Health.

Best regards,

Muthusamy Sivakami

Academic Editor

Thank you for revising the paper based on the comments our reviewer raised.